# Multiple Death Pathways of Neutrophils Regulate Alveolar Macrophage Proliferation

**DOI:** 10.3390/cells11223633

**Published:** 2022-11-16

**Authors:** Xiaochen Gao, Weijia Zhang, Nan Zhang, Qing Yu, Jie Su, Ke Wang, Yanru Chen, Zhen F. Fu, Min Cui

**Affiliations:** 1State Key Laboratory of Agricultural Microbiology, College of Veterinary Medicine, Huazhong Agricultural University, Wuhan 430070, China; 2Key Laboratory of Preventive Veterinary Medicine in Hubei Province, The Cooperative Innovation Center for Sustainable Pig Production, Wuhan 430070, China; 3Key Laboratory of Development of Veterinary Diagnostic Products, Ministry of Agriculture of the People’s Republic of China, Wuhan 430070, China; 4International Research Center for Animal Disease, Ministry of Science and Technology of the People’s Republic of China, Wuhan 430070, China

**Keywords:** alveolar macrophages, neutrophils, influenza virus, proliferation, cell death, inhibitors

## Abstract

Alveolar macrophage (AM) proliferation and self-renewal play an important role in the lung tissue microenvironment. However, the impact of immune cells, especially the neutrophils, on AM homeostasis or function is not well characterized. In this study, we induced in vivo migration of neutrophils into bronchoalveolar lavage (BAL) fluid and lung using CXCL1, and then co-cultured these with AMs in vitro. Neutrophils in the BAL (BAL−neutrophils), rather than neutrophils of bone marrow (BM-neutrophils), were found to inhibit AM proliferation. Analysis of publicly available data showed high heterogeneity of lung neutrophils with distinct molecular signatures of BM− and blood−neutrophils. Unexpectedly, BAL−neutrophils from influenza virus PR8-infected mice (PR8−neutrophils) did not inhibit the proliferation of AMs. Bulk RNA sequencing further revealed that co-culture of AMs with PR8−neutrophils induced IFN-α and -γ responses and inflammatory response, and AMs co-cultured with BAL−neutrophils showed higher expression of metabolism- and ROS-associated genes; in addition, BAL−neutrophils from PR8-infected mice modulated AM polarization and phagocytosis. BAL−neutrophil-mediated suppression of AM proliferation was abrogated by a combination of inhibitors of different neutrophil death pathways. Collectively, our findings suggest that multiple cell death pathways of neutrophils regulate the proliferation of AMs. Targeting neutrophil death may represent a potential therapeutic strategy for improving AM homeostasis during respiratory diseases.

## 1. Introduction

Alveolar macrophages (AMs) are lung tissue-resident plastic cells in the microenvironment of alveolar space. In contrast to other resident macrophages, AMs represent a unique population as the first line of contact with the tissue microenvironment. AMs are sentinels in the pulmonary airspaces and play important roles in the maintenance of homeostasis, tissue repair, and immune surveillance in the respiratory tract [1,2,3]. In the steady state, AMs are required for clearance of dead cells and cellular debris or release of soluble mediators, which avoid host damage and protect pulmonary function [4,5,6,7]. The absence or dysfunction of AMs can result in pulmonary alveolar proteinosis (PAP), a syndrome resulting from abnormal accumulation of alveolar surfactant in the lung [8]; in addition, ablation of AMs was also shown to increase influenza virus-induced lung damage. Therefore, maintenance of AMs is vital for lung homeostasis. In terms of immune surveillance, AMs were shown to suppress T cell proliferation, resulting in the arrest of T cells in G0/G1 phase of the cell cycle [9,10]. AMs have been shown to inhibit pulmonary dendritic cell (DC) maturation and T cell activation [11]. Conversely, AMs were found to promote the generation of Foxp3+ regulating T cells (Foxp3+ Tregs) via presentation of transforming growth factor β (TGF-β) and retinoic acid to CD4 T cells, which improved allergic airway inflammation [12]. However, interferon-γ (IFN-γ) produced by pulmonary T cells during influenza virus infection was shown to inhibit AM-mediated antibacterial defense [13]. AMs may regulate the inflammatory responses of neighboring alveolar epithelial cells (AECs) via cell–cell interaction through cytokines, and in turn, surfactant proteins A and D produced by AECs can also inhibit AM immune responses [14,15,16,17]. Likewise, in another study, the specialized phenotype of lung basophils was found to modulate AM maturation and function by combining IL-33 and GM-CSF of the lung [18]. However, the impact of other immune cell types, especially immune cells in the lung, on AM self-renewal, development, and function is not well understood.

Neutrophils are the first responders of innate immunity, and migrate from bone marrow and circulating blood to the sites of infection, where they phagocytose, kill, or digest the pathogens. However, during primary response, the cytotoxic activity of neutrophils can mediate tissue damage owing to the release of inflammatory cytokines, proteases, and factors contained in the cytoplasmic granules [19]. In contrast, under noninflammatory conditions, neutrophils also play some support functions. For instance, the regulatory mechanisms for maintaining homeostasis of infiltrated neutrophils are important for healthy tissues; however, neutrophils are ultimately eliminated after return to BM or arrival to inflammatory sites, contributing to pathological changes [20,21,22]. Due to the short life span, a large number of dying neutrophils need to be eliminated every day. Usually, the regulation of neutrophil homeostasis is associated with apoptosis of the dying neutrophils. For example, clearance of aging neutrophils in the BM was shown to affect homeostatic rhythmicity in the hematopoietic niche and to trigger hematopoietic progenitor cells into the circulation [23]. In the intestine, tissue macrophages were shown to engulf the infiltrated neutrophils, resulting in enhancement of the BM niche activity [24]. Splenic neutrophils can mediate immunoglobulin class switching and antibody production by activating B cells via B-cell activating factor (BAFF), a proliferation-inducing ligand (APRIL), and IL-21 [25]. Apoptotic neutrophils are engulfed by macrophages and DC phagocytosis, and phagosomes reduce the production of IL-23. Reduced IL-23 inhibits the expression and secretion of IL-17 and G-CSF in γδ T cells and NK T cells. In turn, IL-23 induces IL-17 secretion and stimulates granulopoiesis that leads to the infiltration of circulating neutrophils into tissues [26]. In infection, neutrophil apoptosis was shown to be regulated by Bax/Bak activation (pro-apoptotic BCL-2 family members) to promote M. tuberculosis clearance [27]. The lung infection of P. aeruginosa in NADPH oxidase 2-deficient mice was shown to cause neutrophil pyroptosis depending on caspase-1 activity [28]. On the other hand, influenza virus and severe acute respiratory syndrome coronavirus 2 (SARS-CoV-2) infection led to neutrophil apoptosis and the formation of neutrophil extracellular traps (NETs) [29,30,31].

Both neutrophils and macrophages play important roles in host homeostasis and tissue injury. Among them, NETs of neutrophils promote polarization of AMs toward the M1 phenotype and promote inflammatory response during acute lung injury [32]. Conversely, AMs were found to prevent neutrophil accumulation in lipopolysaccharide (LPS)-induced lung injury [33]. Furthermore, monocyte-derived macrophages can phagocytose NETs and apoptotic cells. However, whether neutrophils regulate AM homeostasis or function is unclear. In our study, neutrophils in the bronchoalveolar lavage fluid (BAL–neutrophils) from healthy mice were found to inhibit AM proliferation. However, neutrophils from the BAL of influenza virus PR8-infected mice (PR8–neutrophils) did not affect the proliferation of AMs. The combination of different cell death inhibitors recovered the proliferation of AMs, including G-CSF (granulocyte-colony-stimulating factor), Q-VD-Oph, NAC (N-acetyl-l-cysteine), and DFO (deferoxamine mesylate), suggesting the involvement of multiple death pathways of neutrophils in maintaining AM homeostasis. This study reveals that neutrophils are important for the maintenance of AM homeostasis, and provides a potential therapeutic approach for lung infectious diseases.

## 2. Materials and Methods

### 2.1. Mouse and Virus

Wild-type C57BL/6 mice (age: 8–12 weeks) were purchased from Laboratory Animal Services of Huazhong Agricultural University and housed in a specific pathogen-free environment at Laboratory Animal Services. All animal experiments were reviewed as per the Guidelines for the Care and Use of Laboratory Animals of the Research Ethics Committee, Huazhong Agricultural University, Hubei, China, and approved by the Research Ethics Committee, Huazhong Agricultural University, Hubei, China (HZAUMO-2019-060).

Influenza A/PR8/34 (PR8) strain was a gift from Prof. Hongbo Zhou of Huazhong Agricultural University. PR8 virus was propagated using 9- to 10-day-old SPF chicken embryos, and viral titers were determined on Madin-Darby canine kidney cells using the plaque-forming unit method. For PR8 strain infection in vivo, PR8 strain (50 PFU/mouse, sublethal)was diluted in Dulbecco’s modified Eagle’s medium (DMEM, Gibco, NY, USA) on ice, and anesthetized mice were inoculated by intranasal route, as described elsewhere [34].

### 2.2. Recruitment of Neutrophils in Bronchoalveolar Lumen

C57BL/6 mice were anesthetized and then intranasally instilled with 5 μg of recombinant murine CXCL1 (rCXCL1, BioLegend, San Diego, CA, USA). Twelve hours after instillation, BAL fluid was collected by flushing the airway five times with 1 mL cold sterile BAL washes ([PBS, Hyclone] with 2% fetal bovine serum [FBS, Gibco] and 2 mM ethylenediaminetetraacetic acid [EDTA, Santa Cruz Biotechnology, Dallas, TX, USA]) via a trachea incision; the obtained fluid was centrifuged at 4 °C, 800× *g* for 5 min. Cell pellet was resuspended by ACK lysis buffer (0.15 M NH4Cl, 1 mM KHCO3, 0.1 mM Na2EDTA, pH 7.2) for 1 min, after centrifugation for 5 min, and then cells were suspended in MACS buffer. To purify the neutrophils, neutrophils were sorted using mouse anti-Ly6G MicroBeads UltraPure (Miltenyi Biotec, Teterow, Germany). Ly6G^+^ cells (BAL–neutrophils) were collected. Cell number was measured by 0.4% Trypan Blue (Hyclone, Logan, UT, USA). For neutrophils of BAL from PR8-infected mice (PR8−neutrophils), mice were sacrificed 3 days post infection, then neutrophils were obtained using anti-Ly6G MicroBeads.

### 2.3. Isolation and Preparation of BM–Neutrophils

BM cells were collected from tibia and femur through a 25-gage needle by flushing with cold, sterile phosphate-buffered saline (PBS). Cells were passed through a 70 μm cell filter in a 50 mL falcon tube and pelleted at 800× *g* for 5 min. After red blood cell lysis, the cell pellet was resuspended in 90 µL of cold MACS buffer. According to the protocol for anti-Ly6G Microbeads (Miltenyi Biotec), Ly6G^+^ cells of BM (BM–neutrophils) were collected by positive magnetic selection using anti-Ly6G beads. The cell number was measured by 0.4% Trypan Blue (Hyclone).

### 2.4. AM Culture In Vitro and Co-Culture with BAL–, BM–, or PR8–Neutrophils

AMs were obtained from BAL as described previously [35]. C57BL/6 mice were sacrificed, and alveolar lavage was performed by flushing the airway for five times with 1 mL cold, sterile BAL washes via a tracheal incision. Alveolar lavages were centrifuged at 4 °C, 800× *g* for 5 min, and cells were obtained and purified to culture plates by adherence in complete medium (RPMI-1640 with 10% FBS and 1% Pen/Strep/glutamate, Thermo Fisher Scientific, MA, USA); 1 × 10^5^ cells per well of co-cultured and mono-cultured AMs were seeded in non-treated 24-well plate for 2 h at 37 °C in 5% CO_2_ cell incubators. The non-adherent cells were washed off with warm PBS and cultured with a complete medium.

For co-culture of AMs with BAL–, BM– or PR8–neutrophils, BAL-, BM- or PR8-neutrophils were plated in a non-treated 24-well plate with AMs (co-cultured at a ratio of 1:1, 2:1, and 8:1). Then the cells were cultured in complete medium containing 10 ng/mL recombinant murine granulocyte-macrophage colony-stimulating factor (GM-CSF, BioLegend, San Diego, CA, USA) after 24 h. Subsequently, the cells were analyzed by flow cytometry or bulk RNA-seq.

### 2.5. BMDM Culture In Vitro and Co-Culture with BAL– or BM–Neutrophils

C57BL/6 mice were sacrificed, and the BM cells were collected from tibia and femur through a 25-gage needle by flushing with cold, sterile PBS. Cells were passed through a 70 μm cell filter in a 50 mL falcon tube and pelleted at 800× *g* for 5 min, and the pellet was suspended in ACK lysis buffer at room temperature (RT). After centrifugation at 800× *g* for 5 min, BM cells were resuspended and cultured with a complete medium (DMEM with 10% FBS and 1% Pen/Strep/glutamate, Thermo Fisher Scientific) supplemented with 50 ng/mL recombinant murine macrophage colony-stimulating factor (M-CSF, BioLegend) to induce differentiation into a macrophage phenotype. After 7 days, BMDMs were generated for subsequent experimentation. For co-culture of BMDMs with BAL– or BM–neutrophils, following replating of BMDMs, 1 × 10^5^ cells per well of co-cultured and mono-cultured BMDMs were plated in a 24-well plate at 37 °C and 5% CO_2_ cell incubators.

### 2.6. Flow Cytometry Analysis

Cell staining was performed with the appropriate antibody cocktail in FACS buffer. The cell subsets were identified based on following cell surface markers: AMs (CD11c^+^Siglec F^+^CD64^+^MerTK^+^), Neutrophils (CD11b^+^Ly6G^+^), BMDMs (F4/80^+^CD11b^+^), macrophage polarization marker (CD80, CD86, CD71 and CD206). Fluorescence-conjugated antibodies CD11b (M1/70), CD11c (N418), CD64 (X54-5/7.1), MerTK (2B10C42), Ly6G (1A8), F4/80 (BM8), CD86 (GL-1), CD80 (16-10A1), CD71 (RI7217) and CD206 (C068C2) were purchased from BioLegend; Ki67 (SolA15) was purchased from Invitrogen; Siglec F (E50-2440) was purchased from (BD Biosciences, Franklin Lakes, NJ, USA). For Ki67 staining, cell suspensions were stained for the surface marker at 4 °C for 30 min in the dark. Cells were washed twice with FACS buffer prior to fixation and permeabilization with the Foxp3 transcription factor staining buffer set (eBioscience, San Diego, CA, USA) for 1 h at RT in the dark. Cells were washed twice with perm wash buffer (eBioscience) and stained with Abs against Ki67 and control immunoglobulin (BioLegend) in perm wash buffer for at least 30 min at RT in the dark. Cells were washed twice with perm wash buffer before samples were processed with the flow cytometer. Samples were collected on flow cytometry and analyzed using FlowJo v10.6.2 software (Tree star) (BD Biosciences, Franklin Lakes, NJ, USA).

### 2.7. Phagocytosis Assay by Flow Cytometry

In accordance with the guidelines for the CellTraceTM CFSE kit (Thermo Fisher Scientific, Waltham, MA, USA), neutrophils were obtained and incubated in a protein-free medium containing Carboxyfluorescein succinimidyl ester (CFSE) for 15 min at RT, followed by wash with a complete medium and quenching of any dye remaining in the solution. Subsequently, BAL– or PR8–neutrophils were added to AM cultures at a ratio of 1:1 for 24 h at 37 °C. The co-culture of AMs and BAL–neutrophils as control. The co-cultured cells were collected with 0.25% Trypsin-EDTA solution and processed for flow cytometry.

### 2.8. Co-Culture Treatment with Different Inhibitors

AMs co-cultured with BAL–neutrophils at a 1:1 ratio were treated with or without different inhibitors in medium, including G-CSF (10 ng/mL, BioLegend), Q-VD-Oph (50 μM, Selleckchem, Houston, TX, USA), NAC (1 mM, Selleckchem), and DFO (1 μM, Selleckchem), the control was co-culture treated without inhibitors. After 24 h, the cells were cultured in complete medium containing 10 ng/mL recombinant murine GM-CSF, and cells were analyzed by flow cytometry.

### 2.9. RNA Sequencing and Analysis

We collected AMs from AMs (alone), co-culture of AMs with BAL–neutrophils, and PR8-neutrophils. Total RNA of AMs was extracted using Trizol kit (Thermo Fisher Scientific) following the manufacturer’s protocol. Two pools per genotype were used for bulk RNA-seq. After quality control, high-quality (Agilent Bioanalyzer RIN of >7.0, Agilent Technologies, Santa Clara, CA, USA) total RNA was used to generate the RNA sequencing library. cDNA synthesis, end-repair, A-base addition, and ligation of the Illumina indexed adapters were performed according to the TruSeq RNA Sample Prep Kit v2 (Illumina, San Diego, CA, USA). The concentration and size distribution of the completed libraries were determined using an Agilent Bioanalyzer DNA 1000 chip (Santa Clara, CA, USA) and Qubit fluorometry (Invitrogen, Carlsbad, CA, USA). Paired-end libraries were sequenced on the DNBSEQ resequencing and PE 150 Kit. Base-calling was performed using DNBSEQ software (DNBSEQ A0, Beijing Genomics Institution, Shenzhen, China).

Paired-end RNA-seq reads were aligned to the mouse reference genome (GRCm38/mm10). Pre- and post-alignment quality controls, gene level raw read count, and normalized read count (i.e., FPKM) were performed using the RSeQC package (v2.3.6)(Beijing Genomics Institution, Shenzhen, China) with the NCBI mouse RefSeq gene model. For functional analysis, Gene set enrichment analysis (GSEA) was performed to identify enriched gene sets using the hallmark collection of the Molecular Signatures Database (MSigDB), containing up- and downregulated genes, and using a weighted enrichment statistic and a log2 ratio metric for ranking genes. Data were submitted to the GEO repository (GSE212080).

### 2.10. Quantitative RT-PCR

The total RNA from the cultured AM as indicated in the text was extracted with Total RNA purification kit (Sigma, Germany) and treated with DNase I (Invitrogen, Waltham, MA, USA). Random primers (Invitrogen) and Moloney murine leukemia virus (M-MLV) reverse transcriptase (Invitrogen) were used to synthesize first-strand cDNA from equivalent amounts of RNA from each sample. qPCR was performed with Fast SYBR Green PCR Master Mix (Applied Biosystems, Waltham, MA, USA). RT-PCR was conducted in duplicates in QuantStudio5 (Applied Bioscience). Results were generated with the comparative threshold cycle (Delta CT) method by normalizing to β-actin.

### 2.11. Statistical Analysis

Unpaired two-tailed Student’s *t*-test (two group comparison), one-way ANOVA (multi-group comparison), Multiple *t*-tests (weight loss and Multiplex studies), or Log-rank (Mantel-Cox) test (survival data) were used to determine statistical significance by GraphPad Prism software (Prism 8, San Diego, CA, USA) and *p*-values < 0.05 were considered indicative of statistical significance.

## 3. Results

### 3.1. BAL–Neutrophils Inhibit AM Proliferation

To investigate the role of neutrophils in AM proliferation or maintenance, we generated a model of neutrophil migration into BAL and lung in C57BL/6 mice induced by administration of recombinant CXCL1 owing to the low number of neutrophils in lung tissues. After 12 h, the percentage of neutrophils in BAL (BAL–neutrophils) reached up to 80% (data not shown), which was consistent with a previous report. Furthermore, the rCXCL1 treatment had no effect on AM and monocyte numbers, and cytokine levels [36]. Purified BAL–neutrophils were co-cultured with AMs at ratios of 1:0, 1:1, 2:1, and 8:1 for 24 h, then the Ki67 (widely used as a biomarker of cell proliferation) expression in AMs was determined with GM-CSF stimulation for 24 h. BAL–neutrophils downregulated Ki67 expression of AMs when co-cultured at a 1:1 ratio (Figure 1A). Neutrophils of bone marrow (BM–neutrophils) were obtained by using anti-Ly6G Microbeads. However, the Ki67 expression in AMs showed no significant alteration after co-culture of BM–neutrophils with AMs at the ratios of 1:0, 1:1, 2:1, and 8:1 (Figure 1B). To explore the difference between these two populations, BAL– and BM–neutrophils, we used a publicly available bulk RNA-seq data set (GSE141745) [20] of neutrophils from lung, blood, and BM of C57BL/6 mice. Heatmap showed differentially expressed genes of neutrophils from lungs compared with neutrophils from BM or blood (Figure 1C). GSEA of hallmark gene sets showed that lung neutrophils increased the expression of genes associated with TNF-α and TGF-β signaling, angiogenesis, P53, and inflammatory responses, but decreased the expression of G2M checkpoint, E2F targets, and mitotic spindle (Figure 1D). Neutrophils from the lung displayed significant enrichment of senescence genes and reduction of differentiation and phagocytosis genes compared to BM–neutrophils (Figure 1E–G). Neutrophils are generated in the bone marrow and released into the circulatory system. Neutrophils are highly motile cells, and during circulation in tissues, they gain the ability for phagocytosis or killing, and finally show aging by losing their proliferative ability [37]. These data revealed that neutrophils in the lung acquired new phenotypes and functional characteristics compared with that from BM and blood, although neutrophils are short-lived in tissues [38,39].

### 3.2. Neutrophils from BAL or BM Do Not Influence BMDM Proliferation

AMs have the ability for self-maintenance independent of the contribution of circulating precursors in situ. However, if AMs are depleted by lethal whole-body irradiation or inflammation, BM-derived monocytes can gain a competitive advantage and readily repopulate the AM niche, which can only differentiate into mature and fully functional AMs in the recipient lungs [40]. To probe whether neutrophils influence the proliferation of BM-derived macrophages (BMDMs), we generated BMDMs by inducing differentiation of BM cells into macrophages using recombinant murine M-CSF. The existence of BAL–neutrophils did not inhibit Ki67 expression of BMDMs (Figure 2A). Similarly, the Ki67 expression of AMs was also not changed in BMDMs co-cultured with neutrophils of BM (Figure 2B). The results suggested that neutrophils, irrespective of whether these were from BAL or BM, only suppressed the proliferation of AMs, but not BMDMs.

### 3.3. PR8−Neutrophils Are Unable to Affect AM Proliferation

AMs are critical for the initial host response to RNA virus infection in the lung, especially influenza virus. To examine the role of neutrophils in AM self-renewal during influenza virus infection, neutrophils from BAL were sorted from PR8 infected mice (PR8–neutrophils) and then co-cultured with AMs at the ratios of 1:0, 1:1, 2:1, and 8:1. PR8–neutrophils did not inhibit Ki67 expression of AMs (Figure 3A). To better understand the potential mechanisms by which BAL– and PR8–neutrophils regulate AM proliferation differently, we analyzed publicly available microarray data (GSE165299) of neutrophils in lung from mice on day 3 post influenza virus infection or from uninfected mice (Figure 3B). GSEA of hallmark gene sets showed that influenza virus infected-neutrophils expressed higher levels of genes related to IFN-α and -γ responses, inflammatory responses, E2F and Myc targets, and metabolism compared with uninfected neutrophils (Figure 3C). Moreover, the gene sets associated with monocyte chemotaxis also showed higher expression (Figure 3D). Nevertheless, most genes expressed in uninfected neutrophils were clustered in transcriptional regulation of granulopoiesis and special markers of neutrophils (Figure 3E). These data showed that the expression of those genes of PR8–neutrophils may cause AM activation, which then possibly lost the function for inhibiting AM proliferation. Combined with the result in Figure 1D, it seems plausible that IFN responses-associated genes or some granule genes released by BAL–neutrophils regulated AM maintenance.

### 3.4. Neutrophils Modulate AM Self-Renewal and Phagocytosis

To further analyze the underlying mechanism affecting AM proliferation, we performed bulk RNA sequencing of mono-cultured AMs and AMs co-cultured with BAL– or PR8–neutrophils. Several differentially expressed genes were found in three groups, especially in AMs co-cultured with PR8–neutrophils (Figure 4A). Comparing AM (mono-cultured AMs) with BAL groups (AMs co-cultured with BAL–neutrophils), hallmark gene sets analysis revealed that AMs co-cultured with BAL–neutrophils tended to show increased expression of genes related to KRAS signaling, xenobiotic metabolism, and inflammatory response, but BAL–neutrophils influenced E2F targets, G2M checkpoint, mitotic spindle, apical junction and myogenesis of AMs (Figure 4B, top). GSEA analysis showed that AMs from PR8 group (AMs co-cultured with PR8–neutrophils) induced IFN-α and -γ responses, TNF-α signaling, and inflammatory response compared with AMs of BAL group. However, AMs of the BAL group showed increased expression of cholesterol homeostasis-associated genes compared with AMs of PR8 group (Figure 4B, bottom). These findings indicate that BAL–neutrophils may help to maintain AM homeostasis. Furthermore, AMs of the BAL group showed decreased expression of cell cycle-associated transcription genes compared with AM alone or AMs of PR8 group (Figure 4C–D), which is consistent with the concept that BAL–neutrophils impair AM proliferation. Gene ontology analysis suggested that neutrophils interrelated chemotaxis and that granule genes were expressed in AMs of the BAL group (Figure 4E). Collectively, these findings suggest that neutrophils of BAL probably preserved homeostasis of AMs even though AM proliferation was impaired.

In this bulk RNA-seq analysis, AMs co-cultured with BAL–neutrophils showed upregulation of xenobiotic metabolism compared to AMs alone. Xenobiotic metabolism pathways are known to be involved in oxidative stress [41]; consistent with this, we found high expression of relative genes of reactive oxygen species (ROS) in AMs co-cultured with BAL–neutrophils (Figure 5A). ROS can activate macrophages, which are usually polarized into classically activated macrophages (M1) and alternatively activated macrophages (M2) [42,43]. M1 macrophages are induced by IFN-γ, LPS, or TNF-α and cause pro-inflammatory responses. Conversely, M2 are triggered by GM-CSF, IL-10 or IL-4, and IL-13, leading to an anti-inflammatory response [44,45]. Subsequently, we assessed whether AMs were activated by neutrophils. We found that mRNA expression of some M1-associated genes in AMs co-cultured with BAL–neutrophils was higher than that in AMs (alone), but PR8-neutrophils caused AMs to highly express M2-associated genes (Figure 5B). Mean fluorescence intensity (MFI) of cell surface marker CD80 and CD86 was higher in AMs co-cultured with BAL–neutrophils, though AMs co-cultured with PR8–neutrophils showed increased expression of CD71 and CD206 (Figure 5C) [46]. This is consistent with the results of other studies in which high GM-CSF level of AMs were shown to induce M1 to M2 switch after influenza A virus infection [47]. In addition, phagocytotic capacity is a classical functional property of macrophages by which they actively engulf unwanted material, dead cells, and debris during maintenance of homeostasis, tissue injury, and infection [48,49]. Usually, M2 macrophages exhibit high phagocytic activity. Herein, the heatmap and Quantitative Realtime-PCR showed that AMs of the PR8 group had more differentially expressed phagocytosis-associated genes than AMs alone or AMs of BAL group (Figure 5D,E). We also measured AM phagocytotic capacity using CFSE-labeled neutrophils in the co-culture system. AMs were found to engulf more CFSE-labeled PR8–neutrophils than BAL–neutrophils (Figure 5F). These data indicate that BAL–neutrophils not only influence AM proliferation, but also affect the phagocytosis.

### 3.5. Cell Death of BAL–Neutrophils Inhibits AM Proliferation

Neutrophils are short-lived immune cells. Spontaneous death of neutrophils may take place at any moment of homeostasis and inflammation, and various cell death pathways are involved, including apoptosis, necrosis, pyroptosis, autophagy, and ferroptosis. For example, aging neutrophils were shown to accumulate ROS, negatively regulate phosphatidylinositol 3,4,5-trisphosphate/Akt signaling and activate pan-caspase-mediated pathways [50,51]. Autophagy regulates degranulation, ROS and inflammatory responses of neutrophils [52]. In this study, AMs and neutrophils were co-cultured for more than 24 h, and neutrophil death may have possibly occurred. Influenza virus infection was shown to result in the apoptosis of neutrophils and the formation of NETs [29,30]; however, Figure 3A suggests that the signaling pathways of apoptosis and NETosis for neutrophils were not major factors for inhibiting AM proliferation. Clinical data has shown that G-CSF is an effective drug to treat neutropenia [53]. In addition, Q-VD-Oph was shown to be a pan-caspase inhibitor of neutrophil death. In a previous study [54], G-CSF, Q-VD-Oph (Quinoline-Val-Asp-Difluorophenoxymethylketone, a pan-caspase inhibitor), NAC (N-acetyl cysteine, an antioxidant mediator), and DFO (deferoxamine mesylate, an iron chelator) were applied alone or in combination to target the neutrophil death pathways. None of the inhibitors when used alone significantly increased Ki67 expression of AMs co-cultured with BAL–neutrophils, but their combined usage increased Ki67 expression of AMs (Figure 6). These data suggested that the cell death of BAL–neutrophils impacted AM proliferation, and that multiple death pathways were involved in this phenomenon.

## 4. Discussion

AM maintenance via proliferation and self-renewal is important for lung tissue development in situ and for host protection during infection. Previous studies have identified a few genes that are required for the development and homeostasis of AMs. GM-CSF and TGF-β were shown to be important for the differentiation of fetal monocytes into AMs and for the postnatal maturation of AMs [55,56]. Furthermore, AMs interact with T cells, DCs, AECs, type 2 innate lymphoid cells (ILC2s), and alveolar type 2 cells (AT2s) via crosstalk between cytokine signaling pathways in order to respond to damage to the lung microenvironment [9,11,17,57,58]. However, the role of neutrophils in AM maintenance is not clear. In the present study, neutrophils isolated from BAL were found to influence AM proliferation, and multiple death pathways of neutrophils were involved in this process.

Neutrophil populations are highly heterogeneous in different tissues, such as lung, blood, BM, spleen, skin, and intestine. Especially in the lung, the neutrophils are endowed with additional biological characteristics and pro-angiogenic functions [20]. Therefore, BAL–neutrophils, but not BM–neutrophils, influence AM proliferation, probably because BAL–neutrophil senescence activates some pathways, such as TGF-β signaling, P53 pathway, or metabolism compared with BM–neutrophils, and those processes may be implicated in neutrophil degranulation. However, neutrophils isolated from BAL of influenza virus-infected mice did not inhibit AM proliferation, but enhanced AM phagocytosis. In addition, AMs of PR8 group showed increased expression of IFN-α and -γ response-associated genes compared with AMs of BAL group. Furthermore, the relative genes of IFN-α and -γ responses were highly expressed in PR8–neutrophils compared with uninfected neutrophils (Figure 3C). These findings collectively suggest that AMs co-cultured with PR8-meutrophils tend to be activated by IFN responses and further regulate AM proliferation and possibly influence AM homeostasis.

However, macrophages also have other phenotypes and functions based on the tissue environment [59]. Firstly, to better understand macrophage physiology, BMDMs are commonly used as an in vitro experimental model for immunological studies [60]. In this study, neutrophils isolated from BAL or BM were not found to affect the proliferation of BMDMs. One potential reason is the high proliferation rate of BMDMs, and that AMs differ from other macrophages with respect to their tissue specificities. However, due to the limitations of studying other macrophages’ isolation and culture, it is unclear whether BAL–neutrophils affect other macrophage proliferation. Hence, we conclude only that BAL–neutrophils only inhibit AM proliferation, but not BMDMs. Secondly, external stimulation can modify macrophage metabolism from the M1 phenotype in the inhibitory state to the M2 phenotype in the repair state [61,62]. In addition, NETs of neutrophils may induce macrophage polarization to M1 phenotype via release of pro-inflammatory cytokines during acute and chronic diseases [32,63]. According to our bulk RNA-seq data, AMs co-cultured with BAL–neutrophils showed increased expression of inflammation-associated genes compared with AM alone. In addition, the levels of M1 markers CD80 and CD86 were higher than those in AM alone. We speculate that cell death participates in regulating AM proliferation during co-culture of AMs and BAL–neutrophils.

Neutrophils play crucial roles in regulating innate and adaptive immune responses in steady state and disease. Usually, the half-life of neutrophils is less than 24 h. Neutrophil death can lead to the release of their cytotoxic components, which may harm the host and cause inflammatory or autoimmune disease. For example, granulocyte transfusion and G-CSF have been used to treat neutropenia induced by bacterial or fungal infections. Nevertheless, this therapeutic option is limited by the rapid death of neutrophils [64]. Global cell depletion of neutrophils following influenza virus infection leads to a more severe inflammatory response and increases disease severity [65]. However, depletion of a small number of neutrophils is conducive to antiviral response [66]. In our study, AMs and neutrophils were co-cultured for more than 24 h in all experiments, and neutrophil death was observed (data not shown). According to Fan et al.’s study [54], we chose some inhibitors to treat co-culture of AMs and BAL–neutrophils. For example, Q-VD-OPh is pan-caspase inhibitor and has an effect on preventing apoptosis involved by caspase 3, 7, 8, 9, 10 and 13 [67,68]. NAC can decrease apoptosis of resting neutrophils [69], and ameliorates neutrophil functions during acute pancreatitis. DFO is a slow iron chelator, which can inhibit neutrophil ROS and NETs [70]. In Figure 6, we found that none of these inhibitors when used alone significantly increased Ki67 expression in AMs in co-culture, but usage of a combination of cell death inhibitors further confirmed the involvement of multiple death pathways in neutrophil cell death. In the steady state, programmed cell death of neutrophils contributes to the maintenance of AM homeostasis. Different signaling pathways participate in the different stages of neutrophil degranulation [71], and BAL–neutrophils are probably in a different degranulation stage from that of BM– and PR8–neutrophils, which may explain the inhibitory effect of BAL–neutrophils on AM proliferation. Notwithstanding, we observed that PR8–neutrophils isolated from the influenza virus infection model lost the inhibitory effect on AM proliferation; however, PR8–neutrophils stimulated AM activation and further impacted AM homeostasis. Thus, targeting neutrophil cell death is a potential therapeutic approach for AM maintenance in some respiratory diseases.

## 5. Conclusions

In summary, neutrophils from different tissues in healthy or infected mice have different characteristics. Our data show that the life and death of neutrophils play an important role in the proliferation and homeostasis of tissue-resident macrophages in pulmonary airspaces, especially with respect to the various modes of death. Combined usage of different inhibitors of neutrophil death may potentially recover AM proliferation and regulate AM homeostasis.

## Figures and Tables

**Figure 1 cells-11-03633-f001:**
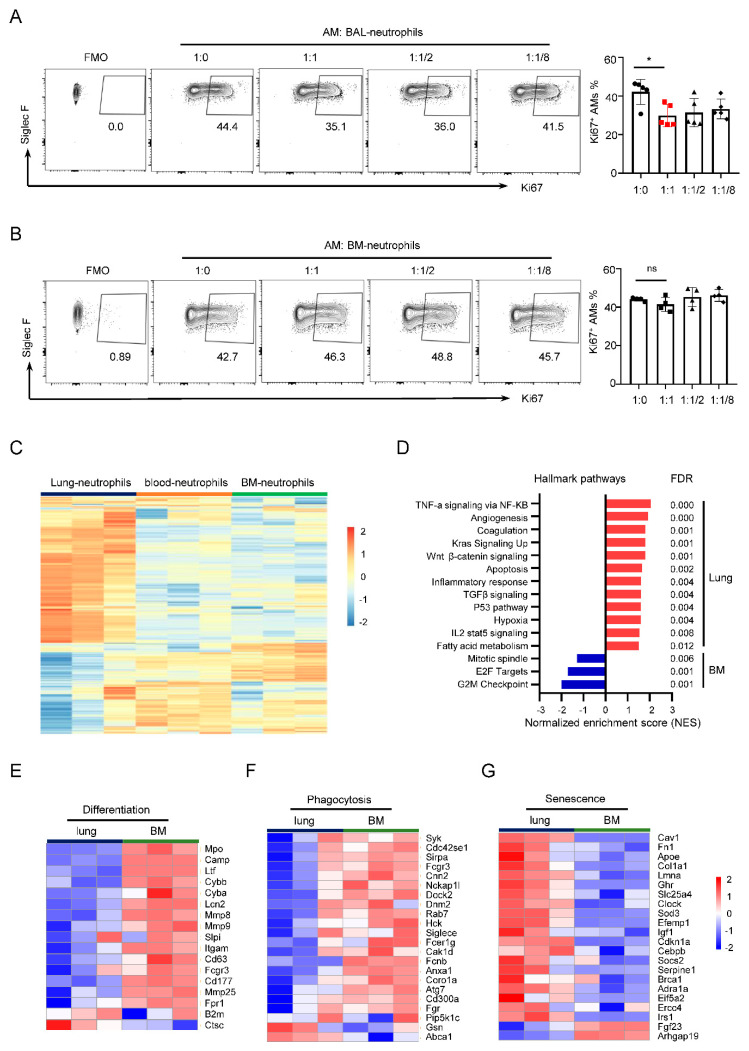
Neutrophils mediate AM proliferation. (**A**) Flow cytometry plots and frequencies of Ki67^+^ AMs in AMs co-cultured with neutrophils isolated from BAL (BAL−neutrophils) at ratios of 1:0, 1:1, 2:1, and 8:1. (**B**) Representative flow cytometry plots and graphs of Ki67^+^ AMs in AMs co-cultured with neutrophils from BM (BM–neutrophils) at same ratios. (**C**) Heatmap showing 300 differentially expressed genes of the top 550 genes in neutrophils of the lung, blood, and BM in vivo from a publicly available bulk RNA-seq dataset (GSE141745). (**D**) Normalized enrichment scores of GSEA from the hallmark gene sets in the molecular signatures database, showing the significantly enriched gene sets in GSE141745. (**E**–**G**) Heatmap showing differentially expressed genes associated with differentiation, phagocytosis, and senescence of neutrophils from GSE141745. Data are presented as arithmetic mean ± SD. ns, no significant; * *p* < 0.05.

**Figure 2 cells-11-03633-f002:**
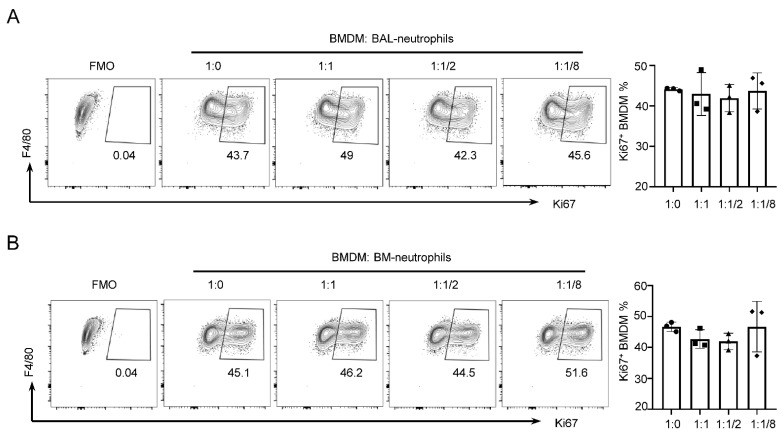
Neutrophils do not affect BMDM proliferation. (**A**) Representative flow cytometry analysis of BMDM proliferation in BMDMs co-cultured with BAL–neutrophils in vitro at a series of ratios. (**B**) Flow cytometry plots and graphs of BMDM proliferation in BMDMs co-cultured with BM–neutrophils in vitro at the same ratios. Data are presented as arithmetic mean ± SD.

**Figure 3 cells-11-03633-f003:**
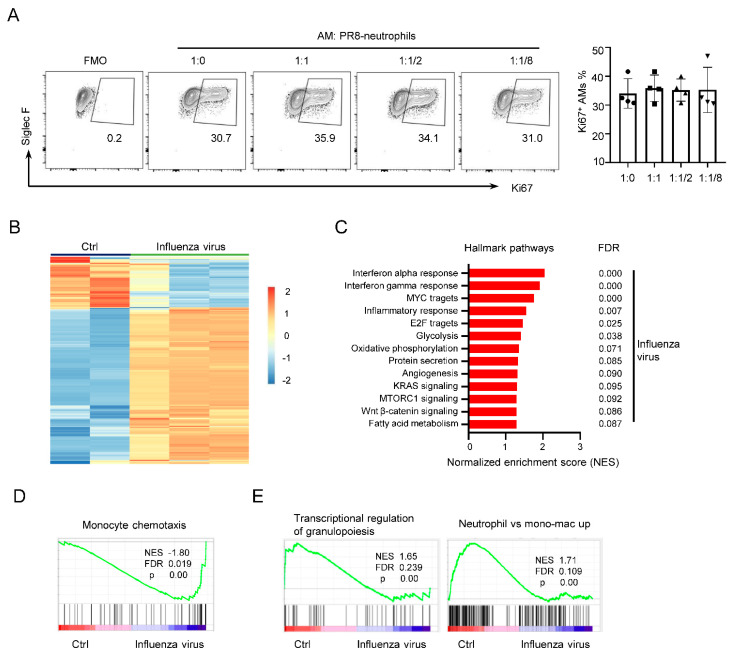
PR8–neutrophils do not inhibit Ki67 expression of AMs. (**A**) Representative flow cytometry plots and frequencies of Ki67^+^ AMs in AMs co-cultured with neutrophils isolated from BAL of PR8 infected mice (PR8–neutrophils) at the indicated ratios. (**B**) Heatmap showing 300 differentially expressed genes out of the top 500 genes in lung neutrophils in publicly available microarray dataset (GSE165299); lung neutrophils were obtained from mice without or with influenza virus at 3 day-post infection. (**C**) Normalized enrichment scores of GSEA from the hallmark gene sets in the molecular signatures database, showing the significantly enriched gene sets in GSE165299. (**D**) Enrichment plots from GSEA of infected neutrophils using gene sets for monocyte chemotaxis shown in GSE165299. (**E**) Enrichment plots from GSEA of uninfected lung neutrophils using gene sets for transcriptional regulation of granulopoiesis and neutrophil vs. mono−mac up shown in GSE165299. Data are presented as arithmetic mean ± SD.

**Figure 4 cells-11-03633-f004:**
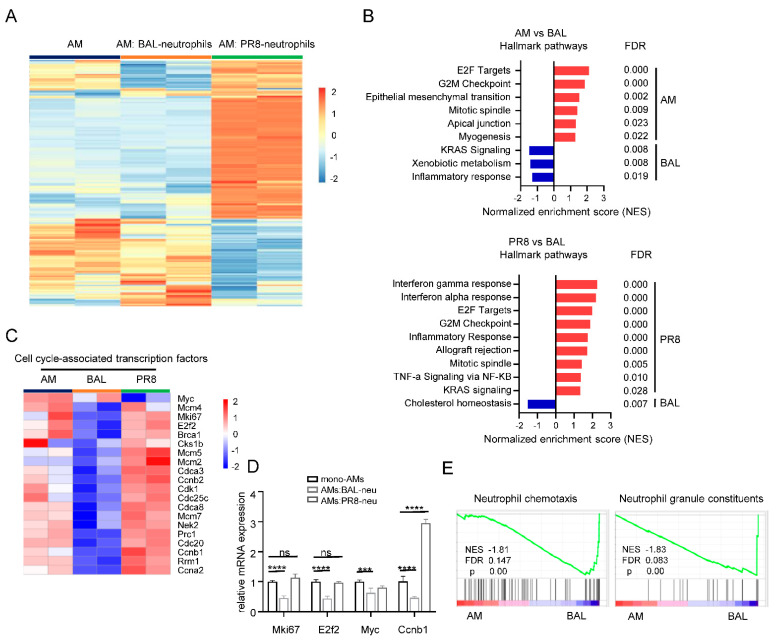
BAL–Neutrophils control AM maintenance. Differentially expressed genes were determined by bulk RNA sequencing of AMs from AMs (alone), AM: BAL–neutrophils (co-culture of AMs and BAL–neutrophils, BAL group), and AM: PR8–neutrophils (co-culture of AMs and PR8–neutrophils, PR8 group) (1:1 ratio). (**A**) Heatmap showing top differentially expressed genes in the three groups. (**B**) GSEA from the hallmark gene sets in the molecular signatures database showing the significantly enriched gene sets between AMs (alone) and AMs of BAL groups, or AMs of PR8 and AMs of BAL groups. (**C**) Heatmap showing differentially expressed genes associated with cell cycle−associated transcription factors from AMs in AMs (alone), AMs of BAL, and AMs of PR8 groups. (**D**) The mRNA of Mki67, E2f2, Myc, and Ccnb1 of AMs from AMs (alone), AM: BAL–neutrophils (AMs: BAL–neu), and AM: PR8–neutrophils (AMs: PR8–neu). (**E**) Enrichment plot from GSEA of infected neutrophils using gene sets for neutrophil chemotaxis and granule constituents between AMs (alone) and AMs of BAL groups. Data are presented as arithmetic mean ± SD. ns, no significant; ***, *p* < 0.001; **** *p* < 0.0001.

**Figure 5 cells-11-03633-f005:**
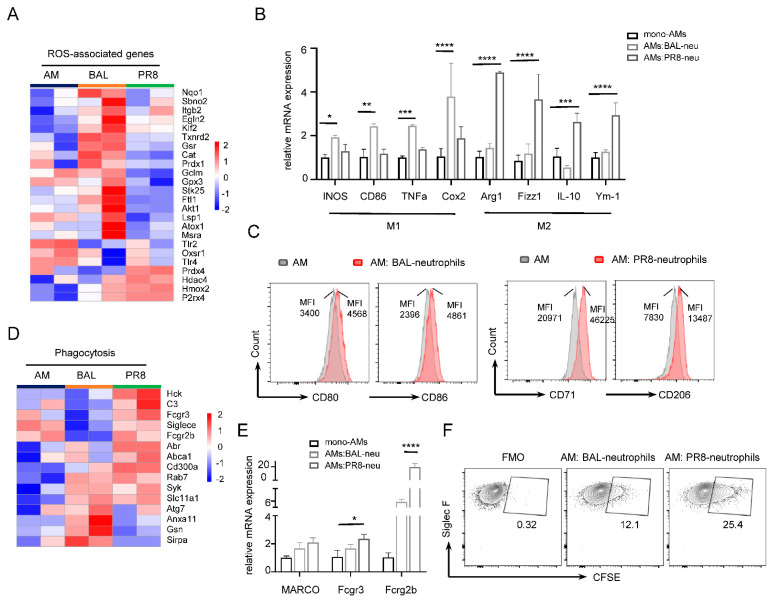
Neutrophils regulate AM polarization and phagocytosis. (**A**) Heatmap showing differentially expressed genes associated with ROS−associated genes in three groups. (**B**) The mRNA expression of M1 and M2−accociated genes of AMs from AMs (alone), AM: BAL–neutrophils (AMs: BAL–neu), and AM: PR8–neutrophils (AMs: PR8–neu). (**C**) Flow cytometry analysis of MFI of CD80, CD86, CD71, and CD206 from AMs co-cultured with BAL–neutrophils or PR8–neutrophils compared to AMs. (**D**) Heatmap showing macrophage−associated phagocytosis genes in AMs from AM (mono-cultured AMs) and BAL groups (co-culture for AMs and BAL–neutrophils), and PR8 (co-culture for AMs and PR8–neutrophils) groups. (**E**) The mRNA of MARCO, Fcgr3, and Fcgr2b of AMs from AMs (alone), AM: BAL–neutrophils (AMs: BAL–neu), and AM: PR8–neutrophils (AMs: PR8–neu). (**F**) Representative flow cytometry plots of CFSE−labeled neutrophils population in gated AMs. Data are presented as arithmetic mean ± SD. * *p* < 0.05; **, *p* < 0.01; ***, *p* < 0.001; **** *p* < 0.0001.

**Figure 6 cells-11-03633-f006:**
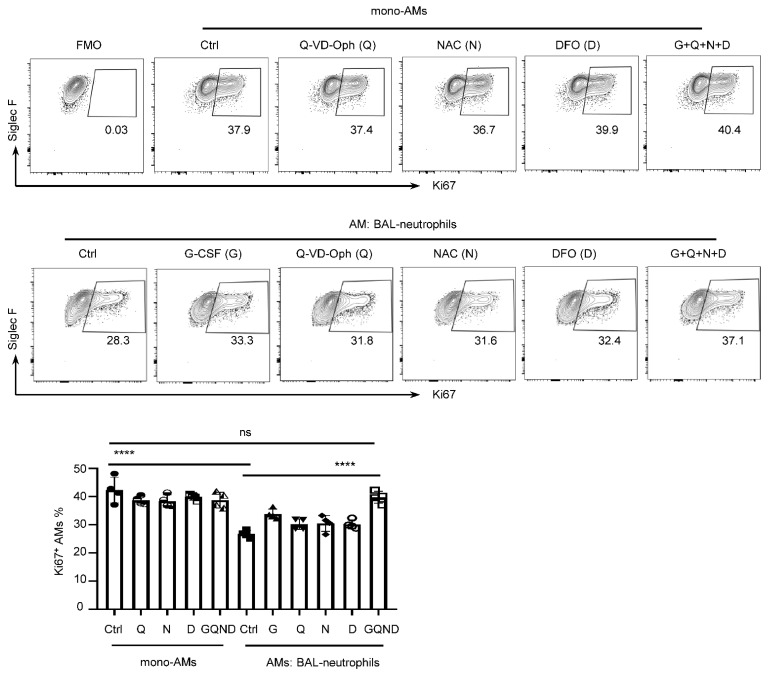
Multiple cell death pathways of BAL−neutrophils affect AM proliferation. Representative flow cytometry plots and frequencies of Ki67^+^ AMs treated with G-CSF (10 ng/mL), Q-VD-Oph (50 μM), NAC (1 mM), DFO (1 μM) in AMs co-cultured with or without BAL−neutrophils at a 1:1 ratio. Data are presented as arithmetic mean ± SD. ns, no significant; **** *p* < 0.0001.

## Data Availability

Data are contained within the article.

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
