# Peer review of "Multiple Death Pathways of Neutrophils Regulate Alveolar Macrophage Proliferation"

_cells, 2022, doi:10.3390/cells11223633_

Round 1
Reviewer 1 Report
To investigate the functional roles of neutrophils in the homeostasis of alveolar macrophage (AM), the authors co-culture of AM with neutrophils from BAL, BM, or PR8-infected BAL, and then the AM proliferation and gene expression were analyzed. They found that co-culture with BAL-neutrophils but not the other two neutrophils can inhibit AM proliferation through multiple cell death pathways. The story is interesting and well-described. I have some minor questions to be answered by the authors.
(1) In figure 6, the bar scales are not consistent with the flow data. For example, the flow of AMs is 37.1%, but the bar is lower than 35%.
(2) In figure 6, mono-AMs treated with drugs might be a better negative control.
(3) The first sentence of the Abstract and Discussion is “Alveolar macrophage (AM) proliferation and self-renewal play an important role in the lung tissue microenvironment. “ May the authors define the difference between “AM’s proliferation” and “AM’s self-renewal?
(4) On Abstract line 26, “BAL-neutrophils modulated AM polarization and phagocytosis,” the BAL-neutrophils should mention from PR8-infected.
(5) On lines 447-448, the authors wrote that neutrophil death was observed in the co-culture of AM-neutrophil. Was the cell death only observed in BAL-neutrophil, not in BM- or PR8-infected neutrophils? This is a key result for the following figure 6’s experiments, so the authors should show the data.
Author Response
Thank you for all comments.

Author Response
Thank you for all comments.

Reviewer 3 Report
This study shows that neutrophils isolated from bronchoalveolar lavage of influenza virus infected mice did not inhibit Alveolar macrophage (AM) proliferation, but enhanced AM phagocytosis. The data is well presented, and the findings seem to be sound.
Author Response
Thank you for all comments.
Reviewer 4 Report
The idea is novel as it is a paper that explains that cell death in neutrophils can affect the proliferation of alveolar macrophages (AM). This study is well-designed to elucidate the effect of neutrophils on the proliferation of AM. However, it did not specifically explain how neutrophil cell death affects the proliferation of AM. The results and discussion could not support the title of this study.
<Major>
118
The authors should explain in more detail the use of the CXCL1 antibody to infiltrate neutrophils with any reference. (or the term antibody was a mistake?)
127
More detailed information on PR8 infection seems to be needed, including references, volume, infection methods, infection route, and virus titers.
128, 133
The authors use the positive selection of neutrophils using anti-Ly6G. Why did you not use the negative selection kit (Milenyi, Cat no 130-097-658)? Is there any possibility that the labeling materials affect the experiment results?
148-153, 228
Why did you use GM-CSF in the co-culture experiment with AM and BAL?
I could not find the detailed methods for AM isolation in the reference to No 35.
325-326
How come to these conclusions?
352-354
A comparison of AM_BAL-neutrophils with AM_PR8-neutrophils for CD80, CD86, CD71, and CD206 is not available in this study. How come to these conclusions?
361-362
It is very confusing whether PR8-neutrophils or BAL-neutrophils increased AM phagocytosis. I think the answers can not be defined based on the results of the study.
381-383
The authors said that PR8-neutrophils do not have AM proliferation inhibition effects. How did the contradictory statements come about?
389-390
It was previously described that AM_BAL-neutrophils suppressed the expression of Ki67. It contradicts the contents of line 389. The expression 'improve' is very confusing.
Figures 1 and 6
In Figure 1, the ratio of Ki67 expression in AM_BAL-neutrophils seems to be 30%. On the other hand, the ratio of that (Ctrl) in Figure 6 appears to be around 20%. Why is there such a difference?
415-417
Is it possible to consider that the cellular activity of TGFβ, P53, and metabolism be one of the pro-angiogenic functions? These pathways are more close to the function of immune responses of neutrophils after infection. A more detailed explanation should be given in this regard.
In addition
The authors should explain in more detail the increase in Ki67 expression in AM_BAL-neutrophils by treatment of inhibitors such as G-CSF, q-VD-Oph, NAC, and DFO. There is no discussion related to this in the Manuscript.
<Minor>
The authors should provide information on centrifuge conditions using the 'x g' unit.
Nationality information must be provided when the reagent company is first introduced.
373:
This is a weird sentence.
Author Response
Thank you for all comments.

Round 2
Reviewer 1 Report
This manuscript has been improved and can be accepted for publication in Cells.
Author Response
Thank you for the comments.
